# Diffusion Approximations for Online Principal Component Estimation and Global Convergence

**Chris Junchi Li**      **Mengdi Wang**      **Han Liu**
Princeton University
Department of Operations Research and Financial Engineering, Princeton, NJ 08544
{junchil,mengdiw,hanliu}@princeton.edu

**Tong Zhang**
Tencent AI Lab
Shennan Ave, Nanshan District, Shenzhen, Guangdong Province 518057, China
tongzhang@tongzhang-ml.org

## Abstract

In this paper, we propose to adopt the diffusion approximation tools to study the dynamics of Oja's iteration which is an online stochastic gradient descent method for the principal component analysis. Oja's iteration maintains a running estimate of the true principal component from streaming data and enjoys less temporal and spatial complexities. We show that the Oja's iteration for the top eigenvector generates a continuous-state discrete-time Markov chain over the unit sphere. We characterize the Oja's iteration in three phases using diffusion approximation and weak convergence tools. Our three-phase analysis further provides a finite-sample error bound for the running estimate, which matches the minimax information lower bound for principal component analysis under the additional assumption of bounded samples.

## 1   Introduction

In the procedure of Principal Component Analysis (PCA) we aim at learning the principal leading eigenvector of the covariance matrix of a $d$-dimensional random vector $\boldsymbol{Z}$ from its independent and identically distributed realizations $\boldsymbol{Z}_1, \ldots, \boldsymbol{Z}_n$. Let $\mathbb{E}[\boldsymbol{Z}] = 0$, and let the eigenvalues of $\boldsymbol{\Sigma}$ be $\lambda_1 > \lambda_2 \geq \cdots \geq \lambda_d > 0$, then the PCA problem can be formulated as minimizing the expectation of a nonconvex function:

$$\begin{aligned} \text{minimize} \quad & -\mathbf{w}^\top \mathbb{E}\left[\boldsymbol{Z}\boldsymbol{Z}^\top\right] \mathbf{w}, \\ \text{subject to} \quad & \|\mathbf{w}\| = 1, \mathbf{w} \in \mathbb{R}^d, \end{aligned} \tag{1.1}$$

where $\|\cdot\|$ denotes the Euclidean norm. Since the *eigengap* $\lambda_1 - \lambda_2$ is nonzero, the solution to (1.1) is unique, denoted by $\mathbf{w}^*$. The classical method of finding the estimator of the first leading eigenvector $\mathbf{w}^*$ can be formulated as the solution to the empirical covariance problem as

$$\widehat{\mathbf{w}}^{(n)} = \operatorname*{argmin}_{\|\mathbf{w}\|=1} -\mathbf{w}^\top \widehat{\boldsymbol{\Sigma}}^{(n)}\mathbf{w}, \qquad \text{where } \widehat{\boldsymbol{\Sigma}}^{(n)} \equiv \frac{1}{n}\sum_{i=1}^{n} \boldsymbol{Z}^{(i)}\left(\boldsymbol{Z}^{(i)}\right)^\top.$$

In words, $\widehat{\boldsymbol{\Sigma}}^{(n)}$ denotes the empirical covariance matrix for the first $n$ samples. The estimator $\widehat{\mathbf{w}}^{(n)}$ produced via this process provides a statistical optimal solution $\widehat{\mathbf{w}}^{(n)}$. Precisely, [43] shows that the angle between any estimator $\widetilde{\mathbf{w}}^{(n)}$ that is a function of the first $n$ samples and $\mathbf{w}^*$ has the following

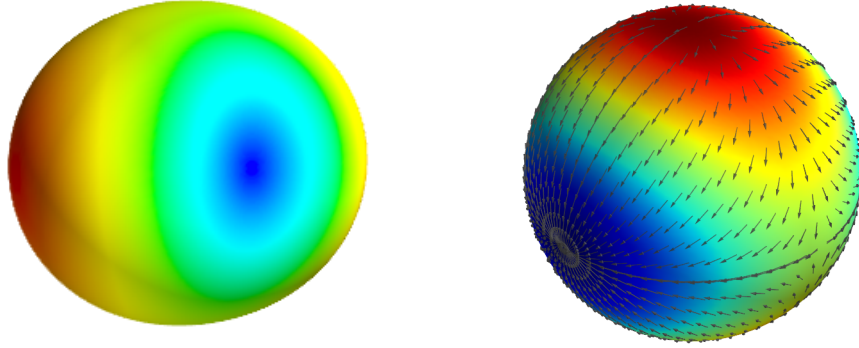

Figure 1: Left: an objective function for the top-1 PCA, where we use both the radius and heatmap to represent the function value at each point of the unit sphere. Right: A quiver plot on the unit sphere denoting the directions of negative gradient of the PCA objective.

*minimax lower bound*

$$\inf_{\widetilde{\mathbf{w}}^{(n)}} \sup_{\boldsymbol{Z} \in \mathcal{M}(\sigma_*^2, d)} \mathbb{E}\left[\sin^2 \angle (\widetilde{\mathbf{w}}^{(n)}, \mathbf{w}^*)\right] \geq c \cdot \sigma_*^2 \cdot \frac{d-1}{n}, \tag{1.2}$$

where $c$ is some positive constant. Here the infimum of $\widetilde{\mathbf{w}}^{(n)}$ is taken over all principal eigenvector estimators, and $\mathcal{M}(\sigma_*^2, d)$ is the collection of all $d$-dimensional subgaussian distributions with mean zero and *eigengap* $\lambda_1 - \lambda_2 > 0$ satisfying $\lambda_1\lambda_2/(\lambda_1 - \lambda_2)^2 \leq \sigma_*^2$. Classical PCA method has time complexity $\mathcal{O}(nd^2)$ and space complexity $\mathcal{O}(d^2)$. The drawback of this method is that, when the data samples are high-dimensional, computing and storage of a large empirical covariance matrix can be costly.

In this paper we concentrate on the *streaming or online method* for PCA that processes online data and estimates the principal component sequentially without explicitly computing and storing the empirical covariance matrix $\widehat{\boldsymbol{\Sigma}}$. Over thirty years ago, Oja [30] proposed an online PCA iteration that can be regarded as a projected stochastic gradient descent method as

$$\mathbf{w}^{(n)} = \Pi\left[\mathbf{w}^{(n-1)} + \beta \boldsymbol{Z}^{(n)}(\boldsymbol{Z}^{(n)})^\top \mathbf{w}^{(n-1)}\right]. \tag{1.3}$$

Here $\beta$ is some positive learning rule or stepsize, and $\Pi$ is defined as $\Pi\mathbf{w} = \|\mathbf{w}\|^{-1}\mathbf{w}$ for each nonzero vector $\mathbf{w}$, namely, $\Pi$ projects any vector onto the unit sphere $\mathcal{S}^{d-1} = \{\mathbf{w} \in \mathbb{R}^d \mid \|\mathbf{w}\| = 1\}$. Oja's iteration enjoys a less expensive time complexity $\mathcal{O}(nd)$ and space complexity $\mathcal{O}(d)$ and thereby has been used as an alternative method for PCA when both the dimension $d$ and number of samples $n$ are large.

In this paper, we adopt the diffusion approximation method to characterize the stochastic algorithm using Markov processes and its differential equation approximations. The diffusion process approximation is a fundamental and powerful analytic tool for analyzing complicated stochastic process. By leveraging the tool of weak convergence, we are able to conduct a heuristic finite-sample analysis of the Oja's iteration and obtain a convergence rate which, by carefully choosing the stepsize $\beta$, matches the PCA minimax information lower bound. Our analysis involves the weak convergence theory for Markov processes [11], which is believed to have a potential for a broader class of stochastic algorithms for nonconvex optimization, such as tensor decomposition, phase retrieval, matrix completion, neural network, etc.

**Our Contributions**     We provide a Markov chain characterization of the stochastic process $\{\mathbf{w}^{(n)}\}$ generated by the Oja's iteration with constant stepsize. We show that upon appropriate scalings, the iterates as a Markov process weakly converges to the solution of an ordinary differential equation system, which is a multi-dimensional analogue to the logistic equations. Also locally around the neighborhood of a stationary point, upon a different scaling the process weakly converges to the multidimensional Ornstein-Uhlenbeck processes. Moreover, we identify from differential equation approximations that the global convergence dynamics of the Oja's iteration has three distinct phases:

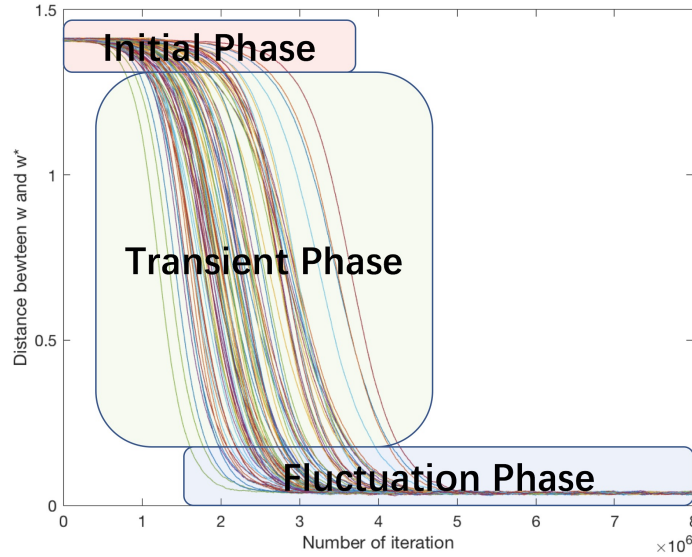

Figure 2: A simulation plot of Oja's method, marked with the three phases.

(i) The initial phase corresponds to escaping from unstable stationary points;

(ii) The second phase corresponds to fast deterministic crossing period;

(iii) The third phase corresponds to stable oscillation around the true principal component.

Lastly, this is the first work that analyze the *global* rate of convergence analysis of Oja's iteration, i.e., the convergence rate does *not* have any initialization requirements.

**Related Literatures**   This paper is a natural companion to paper by the authors' recent work [23] that gives explicit rate analysis using a discrete-time martingale-based approach. In this paper, we provide a much simpler and more insightful heuristic analysis based on diffusion approximation method under the additional assumption of bounded samples.

The idea of stochastic approximation for PCA problem can be traced back to Krasulina [19] published almost fifty years ago. His work proposed an algorithm that is regarded as the stochastic gradient descent method for the Rayleigh quotient. In contrast, Oja's iteration can be regarded as a projected stochastic gradient descent method. The method of using differential equation tools for PCA appeared in the first papers [19, 31] to prove convergence result to the principal component, among which, [31] also analyze the subspace learning for PCA. See also [16, Chap. 1] for a gradient flow dynamical system perspective of Oja's iteration.

The convergence rate analysis of the online PCA iteration has been very few until the recent big data tsunami, when the need to handle massive amounts of data emerges. Recent works by [6, 10, 17, 34] study the convergence of online PCA from different perspectives, and obtain some useful rate results. Our analysis using the tools of diffusion approximations suggests a rate that is sharper than all existing results, and our *global* convergence rate result poses no requirement for initialization.

**More Literatures**   Our work is related to a very recent line of work [3, 13, 21, 33, 38–41] on the global dynamics of nonconvex optimization with statistical structures. These works carefully characterize the *global geometry* of the objective functions, and in special, around the unstable stationary points including saddle points and local maximizers. To solve the optimization problem various algorithms were used, including (stochastic) gradient method with random initialization or noise injection as well as variants of Newton's method. The unstable stationary points can hence be avoided, enabling the global convergence to desirable local minimizers.

Our diffusion process-based characterization of SGD is also related to another line of work [8, 10, 24, 26, 37]. Among them, [10] uses techniques based on martingales in discrete time to quantify the global

convergence of SGD on matrix decomposition problems. In comparison, our techniques are based on Stroock and Varadhan's weak convergence of Markov chains to diffusion processes, which yield the continuous-time dynamics of SGD. The rest of these results mostly focus on analyzing continuous-time dynamics of gradient descent or SGD on convex optimization problems. In comparison, we are the first to characterize the global dynamics for nonconvex statistical optimization. In particular, the first and second phases of our characterization, especially the unstable Ornstein-Uhlenbeck process, are unique to nonconvex problems. Also, it is worth noting that, using the arguments of [26], we can show that the diffusion process-based characterization admits a variational Bayesian interpretation of nonconvex statistical optimization. However, we do not pursue this direction in this paper.

In the mathematical programming and statistics communities, the computational and statistical aspects of PCA are often studied separately. From the statistical perspective, recent developments have focused on estimating principal components for very high-dimensional data. When the data dimension is much larger than the sample size, i.e., $d \gg n$, classical method using decomposition of the empirical convariance matrix produces inconsistent estimates [18, 29]. Sparsity-based methods have been studied, such as the truncated power method studied by [45] and [44]. Other sparsity regularization methods for high dimensional PCA has been studied in [2, 7, 9, 18, 25, 42, 43, 46], etc. Note that in this paper we do not consider the high-dimensional regime and sparsity regularization.

From the computational perspective, power iterations or the Lanczos method are well studied. These iterative methods require performing multiple products between vectors and empirical covariance matrices. Such operation usually involves multiple passes over the data, whose complexity may scale with the eigengap and dimensions [20, 28]. Recently, randomized algorithms have been developed to reduce the computation complexity [12, 35, 36]. A critical trend today is to combine the computational and statistical aspects and to develop algorithmic estimator that admits fast computation as well as good estimation properties. Related literatures include [4, 5, 10, 14, 27].

**Organization** §2 introduces the settings and distributional assumptions. §3 briefly discusses the Oja's iteration from the Markov processes perspective and characterizes that it globally admits ordinary differential equation approximation upon appropriate scaling, and also stochastic differential equation approximation locally in the neighborhood of each stationary point. §4 utilizes the weak convergence results and provides a three-phase argument for the global convergence rate analysis, which is near-optimal for the Oja's iteration. Concluding remarks are provided in §5.

## 2 Settings

In this section, we present the basic settings for the Oja's iteration. The algorithm maintains a running estimate $\mathbf{w}^{(n)}$ of the true principal component $\mathbf{w}^*$, and updates it while receiving streaming samples from exterior data source. We summarize our distributional assumptions.

**Assumption 2.1.** The random vectors $\boldsymbol{Z} \equiv \boldsymbol{Z}^{(1)}, \dots, \boldsymbol{Z}^{(n)} \in \mathbb{R}^d$ are independent and identically distributed and have the following properties:

(i) $\mathbb{E}[\boldsymbol{Z}] = 0$ and $\mathbb{E}\left[\boldsymbol{Z}\boldsymbol{Z}^\top\right] = \boldsymbol{\Sigma}$;

(ii) $\lambda_1 > \lambda_2 \geq \cdots \geq \lambda_d > 0$;

(iii) There is a constant $B$ such that $\|\boldsymbol{Z}\|^2 \leq B$.

For the easiness of presentation, we transform the iterates $\mathbf{w}^{(n)}$ and define the *rescaled samples*, as follows. First we let the eigendecomposition of the covariance matrix be

$$\boldsymbol{\Sigma} = \mathbb{E}\left[\boldsymbol{Z}\boldsymbol{Z}^\top\right] = \mathbf{U}\boldsymbol{\Lambda}\mathbf{U}^\top,$$

where $\boldsymbol{\Lambda} = \mathrm{diag}(\lambda_1, \lambda_2, \dots, \lambda_d)$ is a diagonal matrix with diagonal entries $\lambda_1, \lambda_2, \dots, \lambda_d$, and $\mathbf{U}$ is an orthogonal matrix consisting of column eigenvectors of $\boldsymbol{\Sigma}$. Clearly the first column of $\mathbf{U}$ is equal to the principal component $\mathbf{w}^*$. Note that the diagonal decomposition might not be unique, in which case we work with an arbitrary one. Second, let

$$\boldsymbol{Y}^{(n)} = \mathbf{U}^\top \boldsymbol{Z}^{(n)}, \mathbf{v}^{(n)} = \mathbf{U}^\top \mathbf{w}^{(n)}, \mathbf{v}^* = \mathbf{U}^\top \mathbf{w}^*. \tag{2.1}$$

One can easily verify that

$$\mathbb{E}[\boldsymbol{Y}] = 0, \qquad \mathbb{E}\left[\boldsymbol{Y}\boldsymbol{Y}^\top\right] = \boldsymbol{\Lambda};$$

The principal component of the rescaled random variable $\boldsymbol{Y}$, which we denote by $\mathbf{v}^*$, is equal to $\mathbf{e}_1$, where $\{\mathbf{e}_1, \ldots, \mathbf{e}_d\}$ is the canonical basis of $\mathbb{R}^d$. By applying the orthonormal transformation $\mathbf{U}^\top$ to the stochastic process $\{\mathbf{w}^{(n)}\}$, we obtain an iterative process $\{\mathbf{v}^{(n)} = \mathbf{U}^\top \mathbf{w}^{(n)}\}$ in the rescaled space:

$$
\begin{aligned}
\mathbf{v}^{(n)} = \mathbf{U}^\top \mathbf{w}^{(n)} &= \Pi \left\{ \mathbf{U}^\top \mathbf{w}^{(n-1)} + \beta \mathbf{U}^\top \boldsymbol{Z}^{(n)} \left( \boldsymbol{Z}^{(n)} \right)^\top \mathbf{U} \mathbf{U}^\top \mathbf{w}^{(n-1)} \right\} \\
&= \Pi \left\{ \mathbf{v}^{(n-1)} + \beta \boldsymbol{Y}^{(n)} \left( \boldsymbol{Y}^{(n)} \right)^\top \mathbf{v}^{(n-1)} \right\}.
\end{aligned}
\tag{2.2}
$$

Moreover, the angle processes associated with $\{\mathbf{w}^{(n)}\}$ and $\{\mathbf{v}^{(n)}\}$ are equivalent, i.e.,

$$
\angle(\mathbf{w}^{(n)}, \mathbf{w}^*) = \angle(\mathbf{v}^{(n)}, \mathbf{v}^*).
\tag{2.3}
$$

Therefore it would be sufficient to study the rescaled iteration $\mathbf{v}^{(n)}$ in (2.2) and the transformed iteration $\boldsymbol{Y}^{(n)}$ throughout the rest of this paper.

# 3 A Theory of Diffusion Approximation for PCA

In this section we show that the stochastic iterates generated by the Oja's iteration can be *approximated* by the solution of an ODE system upon appropriate scaling, as long as $\beta$ is small. To work on the approximation we first observe that the iteration $\mathbf{v}^{(n)}$, $n = 0, 1, \ldots$ generated by (2.2) forms a discrete-time, time-homogeneous Markov process that takes values on $\mathcal{S}^{d-1}$. Furthermore, $\mathbf{v}^{(n)}$ holds strong Markov property.

## 3.1 Global ODE Approximation

To state our results on differential equation approximations, let us define a new process, which is obtained by rescaling the time index $n$ according to the stepsize $\beta$

$$
\widetilde{\boldsymbol{V}}^\beta(t) \equiv \mathbf{v}^{\beta, \left( \lfloor t\beta^{-1} \rfloor \right)}.
\tag{3.1}
$$

We add the superscript $\beta$ in the notation to emphasize the dependence of the process on $\beta$. We will show that $\widetilde{\boldsymbol{V}}^\beta(t)$ converges weakly to a deterministic function $\boldsymbol{V}(t)$, as $\beta \to 0^+$.

Furthermore, we can identify the limit $\boldsymbol{V}(t)$ as the closed-form solution to an ODE system. Under Assumption 2.1 and using an infinitesimal generator analysis we have

$$
\left| \widetilde{\boldsymbol{V}}^\beta(t + \beta) - \widetilde{\boldsymbol{V}}^\beta(t) \right| = \mathcal{O}(B\beta).
$$

It follows that, as $\beta \to 0^+$, the infinitesimal conditional variance tends to 0:

$$
\beta^{-1} \mathrm{var} \left[ \widetilde{\boldsymbol{V}}^\beta(t + \beta) - \widetilde{\boldsymbol{V}}^\beta(t) \,\middle|\, \widetilde{\boldsymbol{V}}^\beta(t) = \mathbf{v} \right] = \mathcal{O}(B\beta),
$$

and the infinitesimal mean is

$$
\beta^{-1} \mathbb{E} \left[ \widetilde{\boldsymbol{V}}^\beta(t + \beta) - \widetilde{\boldsymbol{V}}^\beta(t) \,\middle|\, \widetilde{\boldsymbol{V}}^\beta(t) = \mathbf{v} \right] = \left( \boldsymbol{\Lambda} - \boldsymbol{V}^\top \boldsymbol{\Lambda} \boldsymbol{V} \right) \boldsymbol{V} + \mathcal{O}(B^2 \beta^2).
$$

Using the classical weak convergence to diffusion argument [11, Corollary 4.2 in §7.4], we obtain the following result.

**Theorem 3.1.** If $\mathbf{v}^{\beta,(0)}$ converges weakly to some constant vector $\boldsymbol{V}^o \in \mathcal{S}^{d-1}$ as $\beta \to 0^+$ then the Markov process $\mathbf{v}^{\beta, \left( \lfloor t\beta^{-1} \rfloor \right)}$ converges weakly to the solution $\boldsymbol{V} = \boldsymbol{V}(t)$ to the following ordinary differential equation system

$$
\frac{\mathrm{d}\boldsymbol{V}}{\mathrm{d}t} = \left( \boldsymbol{\Lambda} - \boldsymbol{V}^\top \boldsymbol{\Lambda} \boldsymbol{V} \right) \boldsymbol{V},
\tag{3.2}
$$

with initial values $\boldsymbol{V}(0) = \boldsymbol{V}^o$.

We can straightforwardly check for sanity that the solution vector $\boldsymbol{V}(t)$ lies on the unit sphere $\mathcal{S}^{d-1}$, i.e., $\|\boldsymbol{V}(t)\| = 1$ for all $t \geq 0$. Written in coordinates $\boldsymbol{V}(t) = (V_1(t), \ldots, V_d(t))^\top$, the ODE is expressed for $k = 1, \ldots, d$

$$
\frac{\mathrm{d}V_k}{\mathrm{d}t} = V_k \sum_{i=1}^d (\lambda_k - \lambda_i) V_i^2.
$$

One can straightforwardly verify that the solution to (3.2) has

$$V_k(t) = (Z(t))^{-1/2} V_k(0) \exp(\lambda_k t), \tag{3.3}$$

where $Z(t)$ is the normalization function

$$Z(t) = \sum_{i=1}^{d} (V_i^o)^2 \exp(2\lambda_i t).$$

To understand the limit function given by (3.3), we note that in the special case where $\lambda_2 = \cdots = \lambda_d$

$$Z(t) = (V_1^o)^2 \exp(2\lambda_1 t) + \left(1 - (V_1^o)^2\right) \exp(2\lambda_2 t),$$

and

$$(V_1(t))^2 = \frac{(V_1^o)^2 \exp(2\lambda_1 t)}{(V_1^o)^2 \exp(2\lambda_1 t) + \left(1 - (V_1^o)^2\right) \exp(2\lambda_2 t)}. \tag{3.4}$$

This is the formula of the *logistic curve*. Hence analogously, $\boldsymbol{V}(t)$ in (3.3) is namely the *generalized logistic curves*.

## 3.2 Local Approximation by Diffusion Processes

The weak convergence to ODE theorem introduced in §3.1 characterizes the global dynamics of the Oja's iteration. Such approximation explains many behaviors, but neglected the presence of noise that plays a role in the algorithm. In this section we aim at understanding the Oja's iteration via stochastic differential equations (SDE). We refer the readers to [32] for more on basic concepts of SDE.

In this section, we instead show that under some scaling, the process admits an approximation of multidimensional Ornstein-Uhlenbeck process within a neighborhood of each of the unstable stationary points, both stable and unstable. Afterwards, we develop some weak convergence results to give a rough estimate on the rate of convergence of the Oja's iteration. For purposes of illustration and brevity, we restrict ourselves to the case of starting point $\mathbf{v}^{(0)}$ being the stationary point $\mathbf{e}_k$ for some $k = 1, \ldots, d$, and denote an arbitrary vector $\mathbf{x}_{\underline{k}}$ to be a $(d-1)$-dimensional vector that keeps all but the $k$th coordinate of $\mathbf{x}$. Using theory from [11] we conclude the following theorem.

**Theorem 3.2.** Let $k = 1, \ldots, d$ be arbitrary. If $\beta^{-1/2} \mathbf{v}_{\underline{k}}^{\beta,(0)}$ converges weakly to some $\mathbf{U}_{\underline{k}}^o \in \mathbb{R}^{d-1}$ as $\beta \to 0^+$, then the Markov process

$$\beta^{-1/2} \mathbf{v}_{\underline{k}}^{\beta,(\lfloor t\beta^{-1} \rfloor)}$$

converges weakly to the solution of the multidimensional stochastic differential equation

$$\mathrm{d}\boldsymbol{U}_{\underline{k}}(t) = -(\lambda_k \mathbf{I}_{d-1} - \boldsymbol{\Lambda}_{\underline{k}})\boldsymbol{U}_{\underline{k}}\,\mathrm{d}t + \left(\lambda_k \boldsymbol{\Lambda}_{\underline{k}}\right)^{1/2} \mathrm{d}\boldsymbol{B}_{\underline{k}}(t), \tag{3.5}$$

with initial values $\boldsymbol{U}_{\underline{k}}(0) = \mathbf{U}_{\underline{k}}^o$. Here $\boldsymbol{B}_{\underline{k}}(t)$ is a standard $(d-1)$-dimensional Brownian motion. [1]

The solution to (3.5) can be solved explicitly. We let for a matrix $\mathbf{A} \in \mathbb{R}^{n \times n}$ the matrix exponentiation $\exp(\mathbf{A})$ as $\exp(\mathbf{A}) = \sum_{n=0}^{\infty} (1/n!)\mathbf{A}^n$. Also, let $\boldsymbol{\Lambda}^{1/2} = \mathrm{diag}\left(\lambda_1^{1/2}, \ldots, \lambda_d^{1/2}\right)$ for the positive semidefinite diagonal matrix $\boldsymbol{\Lambda} = \mathrm{diag}(\lambda_1, \ldots, \lambda_d)$. The solution to (3.5) is hence

$$\boldsymbol{U}_{\underline{k}}(t) = \exp\left[-t(\lambda_k \mathbf{I}_{d-1} - \boldsymbol{\Lambda}_{\underline{k}})\right] \boldsymbol{U}_{\underline{k}}^o + \left(\lambda_k \boldsymbol{\Lambda}_{\underline{k}}\right)^{1/2} \int_0^t \exp\left[(s-t)(\lambda_k \mathbf{I}_{d-1} - \boldsymbol{\Lambda}_{\underline{k}})\right] \mathrm{d}\boldsymbol{B}_{\underline{k}}(s),$$

which is known as the *multidimensional Ornstein-Uhlenbeck process*, whose behavior depends on the matrix $-(\lambda_k \mathbf{I}_{d-1} - \boldsymbol{\Lambda}_{\underline{k}})$ and is discussed in details in §4.

Before concluding this section, we emphasize that the weak convergence to diffusions results in §3.1 and §3.2 should be distinguished from the convergence of the Oja's iteration. From a random process theoretical perspective, the former one treats the weak convergence of finite dimensional distributions of a sequence of rescaled processes as $\beta$ tends to 0, while the latter one charaterizes the long-time behavior of a single realization of iterates generated by algorithm for a fixed $\beta > 0$.

# 4 Global Three-Phase Analysis of Oja's Iteration

Previously §3.1 and §3.2 develop the tools of weak convergence to diffusion under global and local scalings. In this section, we apply these tools to analyze the dynamics of online PCA iteration in three phases in sequel. For purposes of illustration and brevity, we restrict ourselves to the case of starting point $\mathbf{v}^{(0)}$ that is near a saddle point $\mathbf{e}_k$. Let $A^\beta \lesssim B^\beta$ denotes $\limsup_{\beta\to 0^+} A^\beta / B^\beta \le 1$, a.s., and $A^\beta \asymp B^\beta$ when both $A^\beta \lesssim B^\beta$ and $B^\beta \lesssim A^\beta$ hold.

## 4.1 Phase I: Noise Initialization

In consideration of global convergence, we analyze the initial phase where the iteration starts at a point on or around $\mathcal{S}_e$ and eventually escapes an $\mathcal{O}(1)$-neighborhood of the set
$$\mathcal{S}_e = \left\{ \mathbf{v} \in \mathcal{S}^{d-1} : v_1 = 0 \right\}.$$
When thinking the sphere $\mathcal{S}^{d-1}$ as the globe with $\pm\mathbf{e}_1$ being the north and south poles, $\mathcal{S}_e$ corresponds to the *equator* of the globe. Therefore, all unstable stationary points (including saddle points and local maximizers) lie on the equator $\mathcal{S}_e$.

## 4.2 Phase II: Deterministic Crossing

In Phase II, the iteration escapes from the neighborhood of equator $\mathcal{S}_e$ and converges to a basin of attraction of the local minimizer $\mathbf{v}^*$. From strong Markov property of the Oja's iteration introduced in the beginning of §3, one can *forget* the iteration steps in Phase I and analyze the iteration from the final iterate of Phase I. Suppose we have an initial point $\mathbf{v}^{(0)}$ that satisfies $(v_1^{(0)})^2 \asymp \delta$, where $\delta$ is a fixed constant in $(0, 1/2)$, Theorem 3.1 concludes that the iteration moves in a deterministic pattern and quickly evolves into a small neighborhood of the principal component $\mathbf{e}_1$ such that $(v_1^{(n)})^2 \asymp 1 - \delta$.

## 4.3 Phase III: Convergence to Principal Component

In Phase III, the iteration quickly converges to and fluctuates around the true principal component $\mathbf{v}^* = \mathbf{e}_1$. We start our iteration from a neighborhood around the principal component, where $\mathbf{v}^{(0)}$ has $(v_1^{(0)})^2 = 1 - \delta$. Letting $k = 1$ in (3.5) and taking the limit $t \to \infty$, we have the limit $\mathbb{E}\|\boldsymbol{U}_{\underline{1}}(\infty)\|^2 = \operatorname{tr}\mathbb{E}\left([\boldsymbol{U}_{\underline{1}}(t)\boldsymbol{U}_{\underline{1}}(t)^\top]\right) = (\lambda_1/2)\operatorname{tr}\left(\boldsymbol{\Lambda}_{\underline{1}}(\lambda_1\mathbf{I}_{d-1} - \boldsymbol{\Lambda}_{\underline{1}})^{-1}\right)$. Rescaling the Markov process along with some calculations gives as $n \to \infty$, in very rough sense,

$$\lim_{n\to\infty} \mathbb{E}\sin^2\angle(\mathbf{v}^{(n)}, \mathbf{v}^*) \asymp \beta \cdot \mathbb{E}\|\boldsymbol{U}_{\underline{1}}(\infty)\|^2 = \beta \cdot \frac{\lambda_1}{2}\operatorname{tr}\left(\boldsymbol{\Lambda}_{\underline{1}}(\lambda_1\mathbf{I}_{d-1} - \boldsymbol{\Lambda}_{\underline{1}})^{-1}\right)$$
$$= \beta \cdot \sum_{k=2}^d \frac{\lambda_1\lambda_k}{2(\lambda_1 - \lambda_k)}. \tag{4.1}$$

The above display implies that there will be some nondiminishing fluctuations, variance being proportional to the constant stepsize $\beta$, as time goes to infinity or at stationarity. Therefore in terms of angle, at stationarity the Markov process concentrates within a $\mathcal{O}(\beta^{1/2})$-radius neighborhood of zero.

## 4.4 Crossing Time Estimate

We turn to estimate the running time, namely the *crossing time*, which is the number of iterates required for the iteration to cross the corresponding regions in different phases. We will use the relation $\mathbf{v}^{(n)} \approx \boldsymbol{V}(n\beta)$ to bridge the discrete-time algorithm and its continuous-time approximation.

**Phase I.** For illustrative purposes we only consider the special case where $\mathbf{v}$ is close to $\mathbf{e}_k$ the $k$th coordinate vector, which is a saddle point that has a negative Hessian eigenvalue. In this situation, the SDE (3.5) in terms of the first coordinate $U(t)$ of $\boldsymbol{U}_{\underline{k}}$ reduces to

$$\mathrm{d}U(t) = (\lambda_1 - \lambda_k)U(t)\,\mathrm{d}t + (\lambda_1\lambda_k)^{1/2}\,\mathrm{d}B(t), \tag{4.2}$$

with initial value $U(0) = 0$. Solution to (4.2) is known as *unstable Ornstein-Uhlenbeck process* [1] and can be expressed explicitly in closed-form, as

$$U(t) = W^{\beta}(t) \exp\left((\lambda_1 - \lambda_k)t\right), \quad \text{where } W^{\beta}(t) \equiv (\lambda_1 \lambda_k)^{1/2} \int_0^t \exp\left(-(\lambda_1 - \lambda_k)s\right) \, dB(s).$$

Rescaling the time back to the discrete-time iteration, we let $n = t\beta^{-1}$ and obtain

$$v_1^{(n)} \asymp \beta^{1/2} W^{\beta}(n\beta) \exp\left(\beta(\lambda_1 - \lambda_k)n\right). \tag{4.3}$$

In (4.3), the term $W^{\beta}(n\beta)$ is approximately distributed as $t = n\beta \to \infty$

$$W^{\beta}(n\beta) \asymp \left(\frac{\lambda_1 \lambda_k}{2(\lambda_1 - \lambda_k)}\right)^{1/2} \chi,$$

where $\chi$ stands for a standard normal variable. We have

$$v_1^{(n)} \asymp \beta^{1/2} \left(\frac{\lambda_1 \lambda_k}{2(\lambda_1 - \lambda_k)}\right)^{1/2} \chi \exp\left(\beta(\lambda_1 - \lambda_k)n\right). \tag{4.4}$$

In order to have $(v_1^{(n)})^2 = \delta$ in (4.4), we have as $\beta \to 0^+$ the crossing time is approximately

$$N_1^{\beta} \asymp (\lambda_1 - \lambda_k)^{-1} \beta^{-1} \log\left(\delta |\chi|^{-1}\right) + (\lambda_1 - \lambda_k)^{-1} \beta^{-1} \log\left(\left(\frac{\lambda_1 \lambda_d}{2(\lambda_1 - \lambda_d)}\right)^{-1/2} \beta^{-1/2}\right). \tag{4.5}$$

Therefore we have whenever the smallest eigenvalue $\lambda_d$ is bounded away from 0, then asymptotically $N_1^{\beta} \asymp 0.5 (\lambda_1 - \lambda_k)^{-1} \beta^{-1} \log\left(\beta^{-1}\right)$. This suggests that the noise helps the iteration to move away from $\mathbf{e}_k$ rapidly.

**Phase II.** We turn to estimate the crossing time $N_2^{\beta}$ in Phase II. (3.3) together with simple calculation ensures the existence of a constant $T$, that depends only on $\delta$ such that $V_1^2(T) \geq 1 - \delta$. Furthermore $T$ has the following bounds:

$$(\lambda_1 - \lambda_d)^{-1} \log\left((1 - \delta)/\delta\right) \lesssim T \lesssim (\lambda_1 - \lambda_2)^{-1} \log\left((1 - \delta)/\delta\right). \tag{4.6}$$

Translating back to the timescale of the iteration, it takes asymptotically

$$N_2^{\beta} \lesssim (\lambda_1 - \lambda_2)^{-1} \beta^{-1} \log\left((1 - \delta)/\delta\right)$$

iterates to achieve $(v_1^{(N_2^{\beta})})^2 \geq 1 - \delta$. Theorem 3.1 indicates that when $\beta$ is positively small, the iterates needed for the first coordinate squared to cross from $\delta$ to $1 - \delta$ is $\mathcal{O}(\beta^{-1})$. This is substantiated by simulation results [4] suggesting that the Oja's iteration moves *fast* from the warm initialization.

**Phase III.** To estimate the crossing time $N_3^{\beta}$ or the number of iterates needed in Phase III, we restart our counter and have from the approximation in Theorem 3.2 and (3.5) that

$$\mathbb{E}(v_k^{(n)})^2 = (v_k^{(0)})^2 \exp\left(-2(\lambda_1 - \lambda_k)\beta n\right) + \beta \lambda_1 \lambda_k \int_0^{\beta n} \exp\left(-2(\lambda_1 - \lambda_k)(t - s)\right) \, ds$$

$$= \beta \cdot \sum_{k=2}^{d} \frac{\lambda_1 \lambda_k}{2(\lambda_1 - \lambda_k)} + \sum_{k=2}^{d} \left((v_k^{(0)})^2 - \beta \cdot \frac{\lambda_1 \lambda_k}{2(\lambda_1 - \lambda_k)}\right) \exp\left(-2\beta(\lambda_1 - \lambda_k)n\right)$$

$$\asymp \beta \cdot \sum_{k=2}^{d} \frac{\lambda_1 \lambda_k}{2(\lambda_1 - \lambda_k)} + \delta \exp\left(-2\beta(\lambda_1 - \lambda_2)n\right).$$

In terms of the iterations $\mathbf{v}^{(n)}$, note the relationship $\mathbb{E}\sin^2 \angle(\mathbf{v}, \mathbf{e}_1) = \sum_{k=2}^{d} v_k^2 = 1 - v_1^2$. The end of Phase II implies that $\mathbb{E}\sin^2 \angle(\mathbf{v}^{(0)}, \mathbf{e}_1) = 1 - (v_1^{(0)})^2 = \delta$, and hence by setting

$$\mathbb{E}\sin^2 \angle(\mathbf{v}^{(N_3^{\beta})}, \mathbf{e}_1) = \beta \cdot \sum_{k=2}^{d} \frac{\lambda_1 \lambda_k}{2(\lambda_1 - \lambda_k)} + o(\beta),$$

we conclude that as $\beta \to 0^+$

$$N_3^{\beta} \asymp 0.5(\lambda_1 - \lambda_2)^{-1} \beta^{-1} \log\left(\delta \beta^{-1}\right). \tag{4.7}$$

## 4.5 Finite-Sample Rate Bound

In this subsection we establish the global finite-sample convergence rate using the crossing time estimates in the previous subsection. Starting from $\mathbf{v}^{(0)} = \mathbf{e}_k$ where $k = 2, \ldots, d$ is arbitrary, the global convergence time $N^\beta = N_1^\beta + N_2^\beta + N_3^\beta$ as $\beta \to 0^+$ such that, by choosing $\delta \in (0, 1/2)$ as a small fixed constant,

$$N^\beta \asymp (\lambda_1 - \lambda_2)^{-1} \beta^{-1} \log\left(\beta^{-1}\right),$$

with the following estimation on global convergence rate as in (4.1)

$$\sin^2 \angle(\mathbf{v}^{(N^\beta)}, \mathbf{v}^*) = \beta \cdot \sum_{k=2}^{d} \frac{\lambda_1 \lambda_k}{2(\lambda_1 - \lambda_k)}.$$

Given a fixed number of samples $T$, by choosing $\beta$ as

$$\beta = \bar{\beta}(T) \equiv \frac{\log T}{(\lambda_1 - \lambda_2)T} \tag{4.8}$$

we have $T \asymp (\lambda_1 - \lambda_2)^{-1} \bar{\beta}(T)^{-1} \log\left(\bar{\beta}(T)\right)^{-1} = N^{\bar{\beta}(T)}$. Plugging in $\beta$ as in (4.8) we have, by the angle-preserving property of coordinate transformation (2.3), that

$$\mathbb{E} \sin^2 \angle(\mathbf{w}^{(N^{\bar{\beta}(T)})}, \mathbf{w}^*) = \mathbb{E} \sin^2 \angle(\mathbf{v}^{(N^{\bar{\beta}(T)})}, \mathbf{v}^*) \leq \sum_{k=2}^{d} \frac{\lambda_1 \lambda_k}{2(\lambda_1 - \lambda_k)} \cdot \frac{\log T}{(\lambda_1 - \lambda_2)T}. \tag{4.9}$$

The finite sample bound in (4.9) is sharper than any existing results and matches the information lower bound. Moreover, (4.9) implies that the rate in terms of sine-squared angle is $\sin^2 \angle(\mathbf{w}^{(T)}, \mathbf{w}^*) \leq C \cdot \lambda_1 \lambda_2 / (\lambda_1 - \lambda_2)^2 \cdot d \log T / T$, which matches the minimax information lower bound (up to a $\log T$ factor), see for example, Theorem 3.1 of [43]. Limited by space, details about the rate comparison is provided in the supplementary material.

## 5  Concluding Remarks

We make several concluding remarks on the global convergence rate estimations, as follows.

**Crossing Time Comparison.** From the crossing time estimates in (4.5), (4.6), (4.7) we conclude

(i) As $\beta \to 0^+$ we have $N_2^\beta / N_1^\beta \to 0$. This implies that the algorithm demonstrates the *cutoff* phenomenon which frequently occur in discrete-time Markov processes [22]. In words, the Phase II where the objective value in Rayleigh quotient drops from $1 - \delta$ to $\delta$ is an asymptotically a phase of short time, compared to Phases I and III, so the convergence curve occurs instead of an exponentially decaying curve.

(ii) As $\beta \to 0^+$ we have $N_3^\beta / N_1^\beta \asymp 1$. This suggests that for the high-$d$ case that Phase I of escaping from the equator consumes roughly the same iterations as in Phase III.

To summarize from above, the cold initialization iteration roughly takes twice the number of steps than the warm initialization version which is consistent with the simulation discussions in [31].

**Subspace Learning.** In this work we primarily concentrates on the problem of finding the top-1 eigenvector. It is believed that the problem of finding top-$k$ eigenvectors, a.k.a. the subspace PCA problem, can be analyzed using our approximation methods. This will involve a careful characterization of subspace angles and is hence more complex. We leave this for future investigations.

## Footnotes

[1] The reason we have a $(d-1)$-dimensional Ornstein-Uhlenbeck process is because the objective function of PCA is defined on a $(d-1)$-dimensional manifold $\mathcal{S}^{d-1}$ and has $d-1$ independent variables.

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
