[Supplementary Material]

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

# A Proofs of Auxiliary Results

This section provides the proofs of auxilary Propositions. For brevity, we use the following notations throughout Section A of the Appendix: (i) The $C$'s with subscripts denotes some positive numerical constants; (ii) The $C$, $C'$, $C''$'s (*without* subscripts) are positive numerical constants whose values may change between lines; (iii) The $\mathbf{v} \equiv \mathbf{v}^{(n)}$ and $\boldsymbol{Y} \equiv \boldsymbol{Y}^{(n+1)}$; (iv) For generic function $f(\mathbf{v})$ the $\Delta f(\mathbf{v}) = f(\mathbf{v}^{(n+1)}) - f(\mathbf{v}^{(n)})$.

Also in this section, we let $\mathcal{F}_n = \sigma(\mathbf{v}^{(k)} : k = 0, 1, \ldots, n)$ be the filtration of the algorithm iterates, i.e. the $\sigma$-field generated by the stochastic iterates by $n$.

## A.1 Analysis of Algorithm

To analyze the algorithm from the view of a Markov chain, we need to understand the increments on each coordinate at each step.

**Proposition A.1.** Under Assumption 2.1, for each $k = 1, 2, \ldots, d$ and $n \geq 0$ we have for all $\beta \leq (3B)^{-1}$ the following:

(i) There exists a random variable $Q_k$ with $|Q_k| \leq C_{A.1,1}B^2\beta^2$ almost surely, such that the increment on coordinate $k$ at iterate $n$ $v_k^{(n+1)} - v_k^{(n)}$ can be represented as

$$v_k^{(n+1)} - v_k^{(n)} = \beta\left((\mathbf{v}^{(n)\top}\boldsymbol{Y}^{(n+1)})Y_k^{(n+1)} - v_k^{(n)}(\mathbf{v}^{(n)\top}\boldsymbol{Y}^{(n+1)})^2\right) + Q_k; \quad (A.1)$$

(ii) The increment has the following bound

$$\left|v_k^{(n+1)} - v_k^{(n)}\right| \leq C_{A.1,2}B\beta; \quad (A.2)$$

(iii) There exists a deterministic function $E_{1,k}(\mathbf{v})$ with

$$\sup_{\mathbf{v} \in \mathcal{S}^{d-1}} |E_{1,k}(\mathbf{v})| \leq C_{A.1,1}B^2\beta^2,$$

such that for all $\mathbf{v} \in \mathcal{S}^{d-1}$,

$$\mathbb{E}\left[v_k^{(n+1)} - v_k^{(n)} \mid \mathbf{v}^{(n)} = \mathbf{v}\right] = \beta v_k\left(\lambda_k - \mathbf{v}^\top\boldsymbol{\Lambda}\mathbf{v}\right) + E_{1,k}(\mathbf{v}). \quad (A.3)$$

To prove Proposition A.1 we first come to show

**Lemma A.2.** For each $n \geq 0$

$$\left|\|\mathbf{v} + \beta(\mathbf{v}^\top\boldsymbol{Y})\boldsymbol{Y}\|^{-1} - 1 + \beta(\mathbf{v}^\top\boldsymbol{Y})^2 + \frac{1}{2}\beta^2(\mathbf{v}^\top\boldsymbol{Y})^2\|\boldsymbol{Y}\|^2\right| \leq C_{A.2}\beta^2(\mathbf{v}^\top\boldsymbol{Y})^4.$$

*Proof.* Since

$$\|\mathbf{v} + \beta(\mathbf{v}^\top\boldsymbol{Y})\boldsymbol{Y}\|^{-1} = \left(1 + 2\beta(\mathbf{v}^\top\boldsymbol{Y})^2 + \beta^2(\mathbf{v}^\top\boldsymbol{Y})^2\|\boldsymbol{Y}\|^2\right)^{-1/2}, \quad (A.4)$$

Taylor expansion suggests for $|x| < 1$

$$(1 + x)^{-1/2} = \sum_{n=0}^{\infty}\binom{-\frac{1}{2}}{n}x^n = 1 - \frac{1}{2}x + \frac{3}{8}x^2 - \frac{5}{16}x^3 + \cdots$$

which is an alternating series for $x \in [0, 1)$, whereas the absolute terms approach to 0 monotonically

$$\left|\binom{-\frac{1}{2}}{n+1}x^{n+1}\right| \leq \left|\binom{-\frac{1}{2}}{n}x^n\right|.$$

Hence the error bound gives

$$\left|(1 + x)^{-1/2} - 1 + \frac{1}{2}x\right| \leq \frac{3}{8}x^2, \qquad x \in [0, 1). \quad (A.5)$$

Noting $|\mathbf{v}^\top\boldsymbol{Y}| \leq \|\boldsymbol{Y}\|$ we have for all $\beta$

$$2\beta(\mathbf{v}^\top\boldsymbol{Y})^2 + \beta^2(\mathbf{v}^\top\boldsymbol{Y})^2\|\boldsymbol{Y}\|^2 \leq 2B\beta + B^2\beta^2.$$

The above display is strictly less than 1 when $\beta \le (3B)^{-1}$, and hence (A.5) applies. Combined with (A.4) we have

$$\left| \|\mathbf{v} + \beta(\mathbf{v}^\top \mathbf{Y})\mathbf{Y}\|^{-1} - 1 + \frac{1}{2}\left(2\beta(\mathbf{v}^\top \mathbf{Y})^2 + \beta^2(\mathbf{v}^\top \mathbf{Y})^2\|\mathbf{Y}\|^2\right) \right| \le \frac{3}{8}\left(3\beta(\mathbf{v}^\top \mathbf{Y})^2\right)^2.$$

Noticing $|\mathbf{v}^\top \mathbf{Y}| \le \|\mathbf{Y}\|$, triangle inequality suggests

$$\left| \|\mathbf{v} + \beta(\mathbf{v}^\top \mathbf{Y})\mathbf{Y}\|^{-1} - 1 + \beta(\mathbf{v}^\top \mathbf{Y})^2 \right| \le C\beta^2\|\mathbf{Y}\|^4 \le CB^2\beta^2,$$

completing the proof.

$\square$

*Proof of Proposition A.1.* Setting $Q = \|\mathbf{v} + \beta(\mathbf{v}^\top \mathbf{Y})\mathbf{Y}\|^{-1} - 1 + \beta(\mathbf{v}^\top \mathbf{Y})^2$. Then

$$\begin{aligned}
\Delta v_k &= \|\mathbf{v} + \beta(\mathbf{v}^\top \mathbf{Y})\mathbf{Y}\|^{-1}\left(v_k + \beta\mathbf{v}^\top \mathbf{Y}\, Y_k\right) - v_k \\
&= \left(1 - \beta(\mathbf{v}^\top \mathbf{Y})^2 + Q\right)\left(v_k + \beta\mathbf{v}^\top \mathbf{Y}\, Y_k\right) - v_k \\
&= \beta\left((\mathbf{v}^\top \mathbf{Y})Y_k - v_k(\mathbf{v}^\top \mathbf{Y})^2\right) + Q_k,
\end{aligned}$$

where

$$Q_k = \left(v_k + \beta\mathbf{v}^\top \mathbf{Y}\, Y_k\right)Q - \beta^2(\mathbf{v}^\top \mathbf{Y})^3 Y_k. \tag{A.6}$$

Note the term

$$\beta\left[(\mathbf{v}^\top \mathbf{Y})Y_k - v_k(\mathbf{v}^\top \mathbf{Y})^2\right]$$

is absolutely bounded by $2B\beta$, and taking expectation gives

$$\begin{aligned}
\mathbb{E}\left[(\mathbf{v}^\top \mathbf{Y})Y_k - v_k(\mathbf{v}^\top \mathbf{Y})^2\right] &= v_k\lambda_k - v_k\mathbb{E}(\mathbf{v}^\top \mathbf{Y})^2 \\
&= v_k\lambda_k - v_k\mathbf{v}^\top\mathbb{E}(\mathbf{Y}\mathbf{Y}^\top)\mathbf{v}^\top = v_k\left(\lambda_k - \mathbf{v}^\top\mathbf{\Lambda}\mathbf{v}\right).
\end{aligned}$$

To this stage, we have verified

$$\Delta v_k = \beta\left((\mathbf{v}^\top \mathbf{Y})Y_k - v_k(\mathbf{v}^\top \mathbf{Y})^2\right) + Q_k. \tag{A.7}$$

(A.1) as long as Eqs. (A.2) and (A.3) in Proposition A.1 can be concluded if

$$|Q_k| \le CB^2\beta^2, \tag{A.8}$$

since this implies for $E_{1,k}(\mathbf{v}) = \mathbb{E}Q_k$ we have $|E_{1,k}(\mathbf{v})| \le \mathbb{E}|Q_k| \le CB^2\beta^2$. To conclude (A.8), note that $\beta \le (3B)^{-1}$ and hence

$$\left| v_k + \beta\mathbf{v}^\top \mathbf{Y}\, Y_k \right| \le 1 + \beta B \le \frac{4}{3}.$$

Lemma A.2 implies

$$|Q| \le C_{A.2}\beta^2(\mathbf{v}^\top \mathbf{Y})^4 \le C_{A.2}B^2\beta^2.$$

Therefore the first term on RHS of (A.6) is absolutely bounded by $2C_{A.2}B^2\beta^2$. For the second term in (A.6) we have

$$|\beta^2(\mathbf{v}^\top \mathbf{Y})^3 Y_k| \le B^2\beta^2.$$

We thereby verified (A.8) by taking $C = 2C_{A.2} + 1$, which completes all the proof of Proposition A.1.

$\square$

## A.2 Proof of Theorem 3.1

*Proof.* Let $V_k^\beta(t) = v_k^{\beta,[t\beta^{-1}]}$, the Proposition A.1 implies for $V_k^\beta(t) = \mathbf{v}$ the change for coordinate $k$ at $t = n\beta$ is

$$V_k^\beta(t + \beta) - V_k^\beta(t) = \beta\left((\mathbf{v}^\top \mathbf{Y})Y_k - v_k(\mathbf{v}^\top \mathbf{Y})^2\right) + R_k,$$

where $|R_k| \le CB^2\beta^2$. (A.3) implies that the infinitesimal mean is

$$\begin{aligned}
\frac{\mathrm{d}}{\mathrm{d}t}\mathbb{E}V_k^\beta(t) &= \beta^{-1}\mathbb{E}\left[V_k^\beta(t + \beta) - V_k^\beta(t)\,\big|\,V_k^\beta(t) = \mathbf{v}\right] \\
&= v_k\left(\lambda_k - \mathbf{v}^\top\mathbf{\Lambda}\mathbf{v}\right) + \mathcal{O}(\beta),
\end{aligned}$$

Using (A.2) we can compute the infinitesimal variance

$$\frac{\mathrm{d}}{\mathrm{d}t}\mathbb{E}(V_k^\beta(t) - v_k)^2 = \beta^{-1}\mathbb{E}\left[(V_k^\beta(t+\beta) - V_k^\beta(t))^2 \,\middle|\, V_k^\beta(t) = \mathbf{v}\right]$$
$$\leq \beta^{-1}\cdot C_{A.1,2}^2 B^2\beta^2 \to 0.$$

Let $V_k(t)$ be the solution to ODE system (3.2) with initial values $V_k(0) = v_k^{(0)}$. Applying standard infinitesimal generator argument [11, Corollary 4.2 in Sec. 7.4] one can conclude that as $\beta \to 0^+$, the Markov process $V_k^\beta(t)$ converges weakly to $V_k(t)$.

□

### A.3 Proof of Theorem 3.2

*Proof.* Let $U_i^\beta(t) = \beta^{-0.5}v_i^{\beta,(\lfloor t\beta^{-1}\rfloor)}$. Proposition A.1 implies for $\boldsymbol{V}^\beta(t) = \mathbf{e}_k + \mathcal{O}(\beta^{0.5})$ the change for coordinate $i \neq k$ at $t = n\beta$ is

$$U_i^\beta(t+\beta) - U_i^\beta(t) = \beta^{-0.5}\cdot\beta\left((\mathbf{v}^\top\boldsymbol{Y})Y_i - (\mathbf{v}^\top\boldsymbol{Y})^2 v_i\right) + \mathcal{O}(B^2\beta^{1.5}).$$

Hence (A.3) allows us to compute the infinitesimal mean as

$$\frac{\mathrm{d}}{\mathrm{d}t}\mathbb{E}U_{\underline{k},i}^\beta(t) = \beta^{-1}\mathbb{E}\left[U_{\underline{k},i}^\beta(t+\beta) - U_{\underline{k},i}^\beta(t)\,\middle|\,\boldsymbol{U}_{\underline{k}}^\beta(t) = \boldsymbol{U}\right]$$
$$= (\lambda_i - \lambda_k)\cdot\beta^{0.5}\cdot v_i + \mathcal{O}(\beta^{1.5}) = -(\lambda_k - \lambda_i)\cdot U_{\underline{k},i}^\beta + \mathcal{O}(\beta).$$

Using (A.2) we can compute the infinitesimal variance for coordinates $i, j \neq k$ [2]

$$\frac{\mathrm{d}}{\mathrm{d}t}\mathbb{E}\left[\left(U_{\underline{k},i}^\beta(t+\beta) - U_{\underline{k},i}^\beta(t)\right)\left(U_{\underline{k},j}^\beta(t+\beta) - U_{\underline{k},j}^\beta(t)\right)\,\middle|\,\boldsymbol{U}_{\underline{k}}^\beta(t) = \boldsymbol{U}\right]$$
$$= \beta^{-1}\cdot\beta\left(Y_k^2 Y_i Y_j\right) + \mathcal{O}(\beta) \to \lambda_k^2\lambda_i^2 1_{i\neq j}.$$

Thus by applying infinitesimal generator argument [11, Corollary 4.2 in Sec. 7.4] one can conclude that as $\beta \to 0^+$, the Markov process $\beta^{-1/2}\mathbf{v}_{\underline{k}}^{\beta,(\lfloor t\beta^{-1}\rfloor)}$ converges weakly to $\boldsymbol{U}_{\underline{k}}(t)$.

□

## B Miscellaneous

**Rate in Rayleigh Quotient.** From (4.9), the convergence rate in terms of the angle between $\mathbf{w}^{(N^{\bar\beta(T)})}$ and $\mathbf{a}_1$ is $C\cdot(d\cdot\log T/T)^{1/2}$. Such a rate of convergence is well-known as *nearly optimal* in $T$, as indicated in [43]. In terms of the objective function as Rayleigh quotient, once the Oja's iteration dives into the neighborhood of principal component its distribution is approximately the stationary distribution

$$v_k \sim N\left(0, \frac{\lambda_1\lambda_k}{2(\lambda_1 - \lambda_k)}\beta\right).$$

Let $F(\mathbf{v}) = \lambda_1 - \mathbf{v}^\top\boldsymbol{\Lambda}\mathbf{v} = \sum_{k=2}^d(\lambda_1 - \lambda_k)v_k^2$ denote the objective function. Hence at stationarity

$$\mathbb{E}F(\mathbf{v}^{(T)}) = \beta\cdot\sum_{k=2}^d(\lambda_1 - \lambda_k)\cdot\frac{\lambda_1\lambda_k}{2(\lambda_1 - \lambda_k)} = \beta\cdot\frac{\lambda_1}{2}\sum_{k=2}^d\lambda_k.$$

Hence by choosing $\beta = \bar\beta(T)$ as in (4.8) the iteration is *approximately* at stationarity, and we obtain

$$\mathbb{E}F(\mathbf{v}^{(T)}) \lesssim C\cdot\frac{\lambda_1\sum_{k=1}^d\lambda_k - \lambda_1^2}{2}\cdot\frac{\log T}{(\lambda_1 - \lambda_2)T}. \tag{B.1}$$

The term $\sum_{k=1}^d\lambda_k$ is called the *effective rank* in the PCA literatures. Note the results in (4.9) and (B.1) do *not* include each other and can be used as different measures for convergence rate estimation.

**Sharpest finite-sample error bound.** We summarize all existing rate of convergence results for online PCA in Table 1. In short, our work provides a finer rate that matches the minimax lower

| Algorithm | $\sin^2\angle(\mathbf{w}^{(n)}, \mathbf{w}^*)$ | Optimality |
|---|---|---|
| Minimax rate [43] | $C \cdot \dfrac{\lambda_1\lambda_2 \cdot d}{(\lambda_1 - \lambda_2)^2} \cdot \dfrac{1}{n}$ | Lower bound |
| Alecton [10] | $C \cdot \dfrac{B\lambda_1 \cdot d}{(\lambda_1 - \lambda_2)^2} \cdot \dfrac{1}{n}$ | No |
| Block power method [15, 27] | $C \cdot \dfrac{B\lambda_1^2}{(\lambda_1 - \lambda_2)^3} \cdot \dfrac{1}{n}$ | No |
| Online PCA, Oja [6] | $C \cdot \dfrac{B^2}{(\lambda_1 - \lambda_2)^2} \cdot \dfrac{1}{n}$ | No |
| Online PCA, Oja [34] | $C \cdot \dfrac{B^2 \cdot d}{(\lambda_1 - \lambda_2)^2} \cdot \dfrac{1}{n}$ | No |
| Online PCA, Oja [17] | $C \cdot \dfrac{B\lambda_1}{(\lambda_1 - \lambda_2)^2} \cdot \dfrac{1}{n}$ | Yes |
| Online PCA, Oja (this work) | $C \cdot \dfrac{\lambda_1}{\lambda_1 - \lambda_2} \displaystyle\sum_{k=2}^{d} \dfrac{\lambda_k}{\lambda_1 - \lambda_k} \cdot \dfrac{1}{n}$ | Yes |

Table 1: Comparable results on the convergence rate of online PCA. Note that our result matches the minimax information lower bound [43] in the case where $\lambda_2 = \cdots = \lambda_d$. Our result provides a finer estimate than the minimax lower bound in the more general case where $\lambda_2 \neq \lambda_d$. Note that the constant $C$ hides poly-logarithmic factors of $d$ and $n$.

bound and suggests the necessity of further work on the minimax theory for PCA [7, 43]. Our informal derivation above suggests a finer error bound than the recent work [17], whose optimal rate result depends on sample bound $B$ instead of eigenvalues. Using the same algorithm, our rate of convergence

$$C \cdot \frac{\lambda_1}{\lambda_1 - \lambda_2} \sum_{k=2}^{d} \frac{\lambda_k}{\lambda_1 - \lambda_k} \cdot \frac{1}{N}$$

is faster than any existing results.

**General SDE from equator.** In §4 we only consider the case where the initialization is near a saddle point. For the general case if we start from some initial measure concentrated around $\mathcal{S}_e$, the approximate SDE (4.2) can be similarly found. Let

$$L(\mathbf{v}) = \frac{\mathbf{v}^\top \mathbf{\Lambda} \mathbf{v} - \lambda_1 v_1^2}{1 - v_1^2} = \frac{\mathbf{v}_{\underline{1}}^\top \mathbf{\Lambda}_{\underline{1}} \mathbf{v}_{\underline{1}}}{\mathbf{v}_{\underline{1}}^\top \mathbf{v}_{\underline{1}}}.$$

$L(\mathbf{v}) \in [\lambda_d, \lambda_2]$ can be regarded as a convex combination of $(d-1)$-dimensional vector $(\lambda_2, \ldots, \lambda_d)^\top$ with weights $(v_2^2/v_1^2, \ldots, v_d^2/v_1^2)^\top$. Recall that Theorem 3.1 has $\mathbf{v}^{\beta,(\lfloor t\beta^{-1}\rfloor)} \approx \mathbf{V}(t)$, so we have the following

$$\mathrm{d}U(t) = [\lambda_1 - L(\mathbf{V}(t))]\, U(t)\, \mathrm{d}t + [\lambda_1 \cdot L(\mathbf{V}(t))]^{1/2}\, \mathrm{d}B(t). \tag{B.2}$$

In comparison with (4.2) we replace $\lambda_2$ by the quantity $L(\mathbf{V}(t))$. The coefficient in the drift term of (B.2) is $\lambda_1 - L(\mathbf{V}(t))$ which is no less than $\beta(\lambda_1 - \lambda_2)$. Since the stochastic equation (B.2) is *not* in closed-form, so we are in lack of a theory of a weak convergence to justify a result analogous to Theorem 3.2. This suggests an interesting problem that is left for future research.

**Validity of small step-size approximation.** Our analysis works in the setting when when the stepsizes are infinitesimal small. To justify this, we detail the discussions as follows.

(i) Choosing small stepsize is a common practice in nonconvex optimization SGD.

   This has many practical reasons. One of the main reason is that if the stepsize is large, a warm-initialized iteration risks bouncing back to the cold region, while the small stepsize guarantee the stability of SGD algorithm and ensure a decrease of function value in the long run.

(ii) The probability of failure is positively correlated to the stepsize.

We choose the stepsize to be (approximately) inversely proportional to $N$ so the convergence rate result holds with high probability when sample $N$ is large. The differential equation approximation method is then very meaningful as it explicitly characterizes what in essence happens in the algorithm iterations.