[Reviews · NeurIPS 2017]

Reviewer 1



This work analyses the classical Oja iteration for finding the first principal component. They show it can be approximated using a stochastic process and provide global convergence guarantees which match the optimal . Although the related work section is very dense, it is well organized and serves as a good starting point for the non-expert reader. As a non expert in this area I find it quite hard to judge the merits of this paper in detail. Having said that it seems that providing the first global convergence guarantees for Oja’s rule ought to be of interest. The analysis seems to thoroughly explain the different phases in the optimization procedure which perhaps could be useful as a diagnostic tool. The analysis only considers the standard Oja rule which obtains the leading eigenvector. This is perhaps of limited use, but I appreciate the difficulty in extending this to the entire prinicpal subspace. Line 27. A point which is tangential to the direction of a paper but still I believe a mischaracterization: We don’t need to compute the covariance matrix. Computing the SVD suffices and can be sped up using randomized techniques. At the very least, we don’t require O(d^2) storage. There are plenty of typos and ungrammatical sentences throughout and should be proofread more thoroughly. E.g. 145: using an infinitesimal generator analysis results 160: is name the generalized logistic curves. 165: for more on basic concepts and relates topics of SDE 171: a local maximizers that has negative curvatures in every direction. 171: In below, we consider

Reviewer 2



The paper analyzes a streaming algorithm for PCA, proves that the bounds achieved match the minimax rates and breaks the convergence speed of the algorithm in 3 phases thanks to the use of differential equations that capture the evolution state of the recursive power-method algorithm. The presentation of the somehow complex idea behind writing a ODE which captures the convergence of the algorithm is very clear. I wish the authors had reported the result stated at l.148 in the appendix. Finding the reference is not easy. I believe that using the same machinery for analyzing convergence of other nonconvex problems can lead to useful results on problems where the computational algorithms are hard to analyze with standard tools. Can the authors discuss this point in the concluding remarks section?

Reviewer 3



This is a fairly well written paper. The goal of the paper is to study the dynamics of Oja's online algorithm for the estimation of the principal component in PCA. The innovative aspects of the method lie in that the author(s) resort to using a Markov process and via diffusion approximation arguments end up to an equivalent differential equation. In the sequel, they employ weak convergence arguments for Markov processes to reach their results. In this way, assumptions concerning initialization requirements are bypassed. Instead, a global rate analysis philosophy is adopted. The obtained results are interesting in that they show that the algorithm has the potential to escape for unstable stationary points and finally the estimate oscillates around the true principle component. Once the problem is stated, the paper follows well formed arguments. I did not checked all the details of the proof, however, the proof evolves in a mathematically sound way. My main criticism of the paper is that it would be helpful to the readers and beneficial to the paper to have some related simulation results and performance figures.